# Evaluating accurate and efficient sampling strategies designed to measure social behavior and brush use in drylot housed cattle

Claudia Carolina Lozada[1], Rachel M. Park[2], Courtney L. Daigle[1]*

1 Department of Animal Science, Texas A&M University, College Station, Texas, United States of America,
2 Department of Clinical Sciences, College of Veterinary Sciences, North Carolina State University, Raleigh, North Carolina, United States of America

* cdaigle@tamu.edu

## Abstract

Efficient sampling strategies expedite behavioral data collection. While multiple studies have evaluated sampling strategies for core behaviors in cattle, few have focused on social interactions. To identify sampling strategies that accurately captured cattle social behaviors and brush use feedlot steers (n = 3 pens; 9 steers/pen) were observed from 8:00 to 17:00. Average bout duration (sec), total duration per day (sec), and bout frequency were recorded for allogrooming, bar licking, tongue rolling, and brush utilization. Frequency was recorded for headbutting and mounting. Data was extracted from continuous observation datasets using eight different sampling strategies and the results subsequently compared. Differences among sampling strategies were evaluated using a non-parametric One-Way ANOVA Kruskal-Wallis Test. Pearson correlation evaluated the strength of association between a specific sampling strategy and continuous observations. Bout duration for allogrooming ($P > 0.65$), bar licking ($P > 0.60$), tongue rolling ($P > 0.99$), brush use ($P > 0.99$), and mounting frequency ($P > 0.70$) did not differ from continuous observations. Tongue rolling ($r^2 > 0.95$, $P < 0.0001$) and brush use ($r^2 > 0.70$, $P < 0.0003$) were best captured when cattle were observed from 08:00 to 14:00. When cattle were continuously observed from 08:00 to 14:00 or for 15 minutes every 30 minutes, allogrooming ($P > 0.2$) (frequency, duration), bar licking ($P > 0.95$) (frequency, duration), brush use ($P > 0.1$) (frequency, duration), heat butt ($P > 0.30$) (frequency), or tongue rolling ($P > 0.30$) (frequency, duration) did not differ from continuous observations. Observing cattle for 15 minutes every 30 minutes yielded the highest accuracy for all behavioral metrics and was considered the most effective strategy for comprehensively evaluating cattle social behavior ($r^2 > 75$; $P < 0.05$). These results provide insight into accurate and efficient sampling strategies that expedite social behavior data collection in cattle and will facilitate efficient generation of new knowledge regarding cattle social behaviors.

**Data Availability Statement:** All relevant data are within the paper and its Supporting Information files.

**Funding:** The authors received no specific founding for this work.

**Competing interests:** The authors have declared that no competing interest exist.

## Introduction

Cattle are social animals, and as such, the performance of social behaviors (or lack thereof) can provide insight into an animal's welfare state [1]. Cattle are intelligent and curious creatures; thus, social interactions are a critical component of their ethos. Social interactions are a requirement of group living and can have a positive or negative valence. Animals housed in groups perform affiliative behaviors (e.g., allogrooming) that are associated with positive emotions [2], have calming effects [3], facilitate the formation of social bonds [4], and can result in improved coat hygiene [2]. However, living in groups is also accompanied with inherent costs, including competition for resources [5] such as food, water, and mechanical brushes [6]. If resources are scarce, highly valued, or defended, animals may engage in agonistic interactions, which could result in injury or death. For example, in overstocked pens, cattle tend to perform more agonistic interactions, and the cattle that are less successful at displacing spend more time lying down, which affects the pen dynamic [7]. Thus, the frequency, duration, and circadian pattern of social behaviors can provide feedback to producers regarding resource availability, herd synchrony, and level of psychosocial stress. An animal's response to a stressor is related to the characteristics of the stressor, such as predictability and controllability, in addition to characteristics of the individual experiencing the situation, such as coping style, genetics, sex, and life experiences [8].

Living in captivity occasionally results in the development of behaviors that differ from the behavioral repertoire of the species' wild counterpart [9, 10]. In some cases, these behaviors manifest as stereotypies and, as such, consist of repeated movements that seem to lack any function in the context in which they are performed [9]. While the development of stereotypic behaviors can be indicative of an individual having difficulty coping with their current conditions, the persistence of these behaviors may be indicative of a positive welfare state. The performance of stereotypic behaviors can be rewarding and a self-reinforcing strategy to cope with their current scenario [10]. Cattle typically stop performing stereotypies (e.g., bar licking and tongue rolling) when allowed to graze, but they will resume high levels of stereotypies after re-tethering post grazing [11] due to the fact that that diet is a key factor affecting stereotypic behavior [12].

Two primary sampling strategies are typically used to capture animal behavior: continuous recording and scan sampling [13]. Continuous sampling is a true record of the animal's behavior, as relevant behaviors are decoded for the entire duration of time. Continuous sampling provides the most accurate representation of a group or individual's behavioral repertoire [14] and can capture behaviors that occur infrequently, of short duration, or on a circadian pattern [15]. However, this method is labor intensive and time consuming [16]. With the onset of precision livestock management, and the accompanying challenges regarding processing large data sets and preserving sensor technology battery life, identifying alternative sampling strategies that provide an accurate representation of the continuous record has ethological and technological implications [17].

Behavior is an objective measurement that can inform welfare assessment and cattle management. Multiple studies have evaluated sampling strategies designed to expedite data collection regarding lying, standing, and brush use behaviors [16, 18]. However, these studies have varied in their implementation of continuous observations, and ultimate recommendations have included identifying context-specific sampling strategies [19]. Consequently, there are no current recommendations for behavioral sampling strategies to ascertain information about social and stereotypical behaveior of drylot-housed cattle. Thus, the objective of this study was to identify a sampling strategy that could accurately and efficiently record the performance of social, agonistic, and stereotypic behaviors in drylot-housed cattle with access to brushes.

## Materials and methods

Data for the present study were derived from a portion of a larger data set. The methodology was previously outlined in Park et al. [14] and is briefly described below. All procedures for this research were approved by the West Texas A&M—Cooperative Research, Education and Extension Team University Animal Care and Use Committee (approval number 01−09−17). The experimental period lasted 253 days (November 2017—July 2018) with the day of brush implementation serving as d 0. Cattle arrived at the feedlot on d -55 and were slaughtered on d 161 and d 198, respective to their weight block.

### Housing, diet, and treatments

Fifty-four predominately British and British-Continental crossbred steers were shipped to the Texas A&M Agrilife Research Feedlot in Bushland, Texas, United States, in the fall of 2017. Weather conditions are outlined in Fig 1. Cattle were blocked by weight into a light (283.95+/-13.75 kg) and a heavy block (320.69+/- 12.97 kg) before being allocated to one of six pens. Three of these pens had a stationary L-shaped brush. For this research, only the pens with brushes were included in the analysis. Each pen was 25.5 m by 7 m (19.83m$^2$ per head) with earthen flooring. Shade was provided in the form of a partial roof covering (5 x 7 m; 5m$^2$ per head). Each pen provided nine individual Calan head gate feeders and housed nine animals accordingly. Water was provided *ad libitum* from an automatic water trough. A schematic of the pen layout is included in Park et al. [14].

### Behavioral observations

Cattle behavior in the pen was recorded from 08:00 to 17:00 on d 1, 16, and 64, relative to brush placement implementation using a closed-circuit video camera recording system manufactures by Safesky, which was installed to ensure there were no blind spots within the pen. Behavioral data was decoded from video recordings using the continuous sampling method [13, 18]. The data focused on the frequency and duration each steer spent engaged in allo-grooming, bar licking, tongue rolling, and utilizing the brush, as well as the frequency of head-butting and mounting. All continuous behavioral data were collected by 23 trained observers

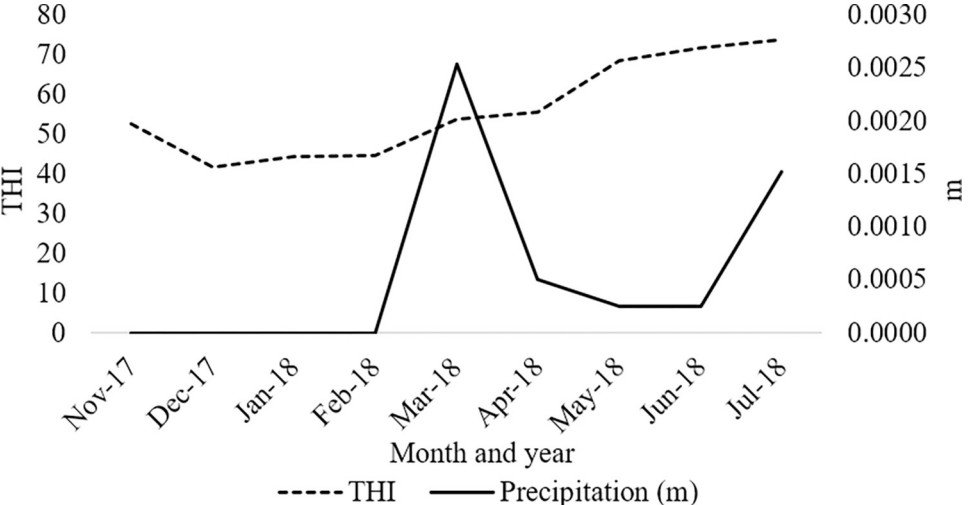

**Fig 1. THI and average precipitations (in) per month in Bushland, TX from November 2017 to July 2018.** Data was retrieved from https://www.wunderground.com/history/daily/us/tx/bushland/KTXBUSHL6/date/2022-8-1.

**Table 1. Sampling strategy description, acronym, and total duration of time (min) of video that would need to be evaluated to collect the behavioral data.**

| Interval ID | Description | Total duration of time/day evaluated (min) |
|---|---|---|
| Continuous | Video was decoded from 08:00 to 20:00 | 720 |
| 5EV30 | 5 minutes of video was decoded every 30 minutes | 70 |
| 5EV60 | 5 minutes of video was decoded every 60 minutes | 35 |
| 10EV30 | 10 minutes of video was decoded every 30 minutes | 140 |
| 10EV60 | 10 minutes of video was decoded every 60 minutes | 70 |
| 15EV30 | 15 minutes of video was decoded every 30 minutes | 210 |
| 15EV60 | 15 minutes of video was decoded every 60 minutes | 105 |
| 8to14 | Video was decoded from 8:00 to 14:00 | 360 |
| 14to16 | Video was decoded from 14:00 to 16:00 | 120 |

utilizing BORIS (Version 6.1.4 [20]). Interobserver reliability between observers and trainer, as well as among observers, was no less than 95% accuracy.

## Sampling strategies

Eight different sampling strategies were selected (Table 1) to evaluate cattle brush use, social and stereotypical behavior. Each of these sampling strategies was extracted from the continuous observation data.

## Statistical analysis

Data were extracted from the continuous observation data set according to eight different sampling strategies (Table 1) to evaluate the impact of sampling strategy on overall outcome. Average duration of time per bout (sec/bout), number of bouts per day (count/d), and total duration of time per day (sec/d) spent performing each behavior were calculated for each sampling strategy. Normality was evaluated using the univariate procedure in SAS (SAS University Edition). According to the Anderson-Darling and Kolmogorov-Smirnov tests, behavioral values for allogrooming, headbutt, tongue rolling, bar licking, and mount and brush use were not normal and could not be normalized. To test for significant differences among the sampling strategies for each behavior, a Kruskal-Wallis Test was evaluated with a nonparametric One-Way ANOVA. To evaluate the strength of association between the results generated with a specific sampling strategy against the results generated using continuous observation, a Pearson correlation (PROC CORR) was used to correlate the results from the average duration of bout, number of bouts, and total duration of all bouts for individual steers with the results of continuous observation using PROC CORR in SAS software (SAS University Edition).

## Results

### Allogrooming

While the average duration of an allogrooming bout did not differ across sampling strategies (Fig 2A), two sampling strategies differed from continuous observations for the frequency (Fig 2B) and total duration of time spent allogrooming per day (Fig 2C). Outcomes from the sampling strategies 5EV60 and 5EV30 differed from continuous observations regarding the frequency of allogrooming bouts ($P > 0.10$) and the total duration ($P > 0.21$) of time spent per day engaged in allogrooming behavior. The sampling strategy 15EV30 had the greatest strength of association with continuous observations ($r^2 > 0.80$; $P < 0.0001$), while the sampling strategy 5EV60 ($r^2 > 0.65$; $P < 0.06$) had the weakest relationship (Fig 3A).

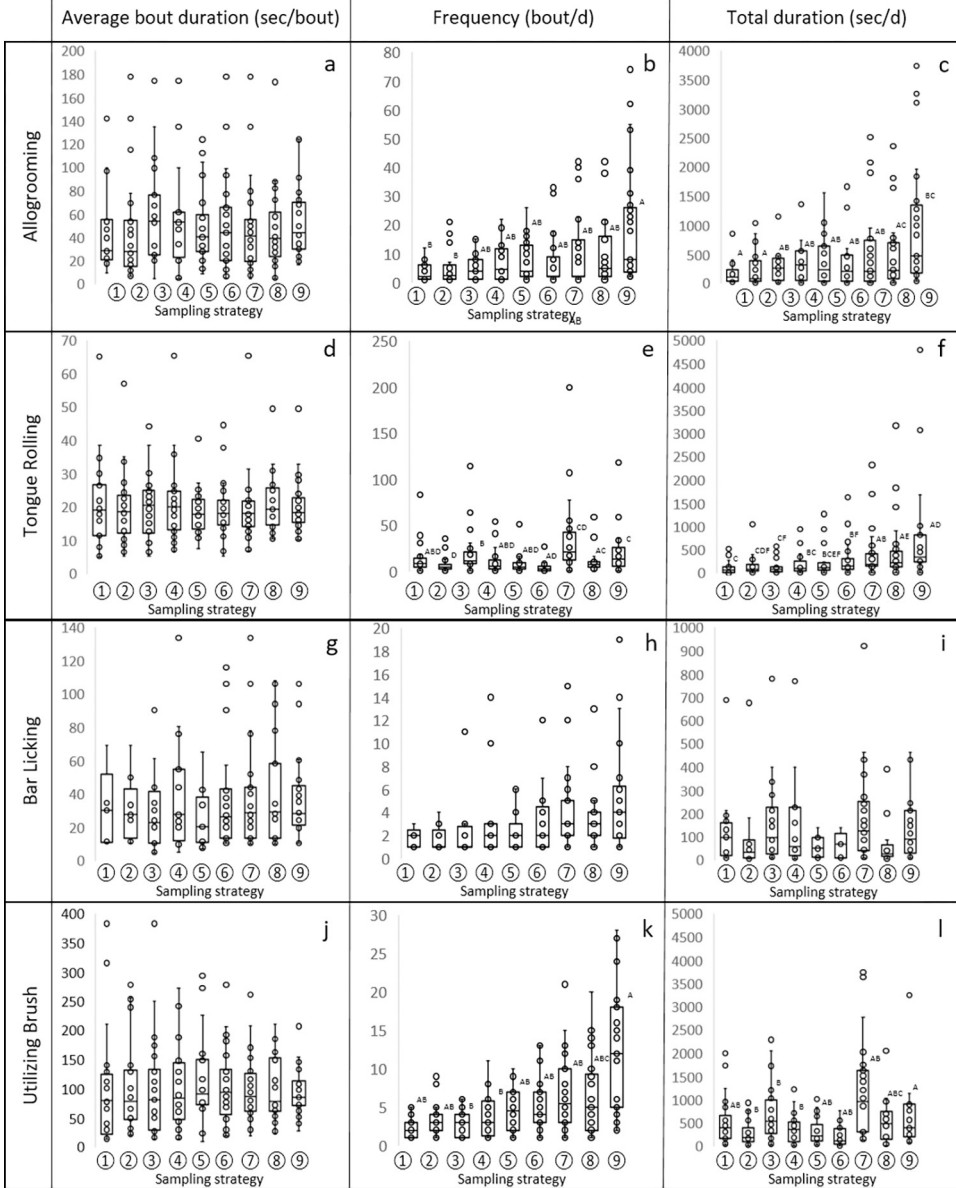

**Sampling Strategy Legend:** ① = Five minutes decoded every 60 minutes (5EV60); ② = Five minutes decoded every 30 minutes (5EV30); ③ = Ten minutes decoded every 60 minutes (10EV60); ④ = Fifteen minutes decoded every 60 minutes (15EV60); ⑤ = Decoded from 14:00 to16:00 (14to16); ⑥ = Ten minutes decoded every 30 minutes (10EV30); ⑦ = Fifteen minutes decoded every 30 minutes (15EV30); ⑧ = Decoded from 08:00 to 14:00 (8to14); ⑨ = Continuous

**Fig 2. Impact of sampling strategy on drylot-housed steer behavior.** Steer behaviors were decoded from video recordings for: a) allogrooming bout duration (sec/bout), b) allogrooming bout frequency (bout/d), c) allogrooming total duration (sec/d), d) bar licking bout duration (sec/bout), e) bar licking bout frequency (bout/d), f) bar licking total duration (sec/d), g) tongue rolling bout duration (sec/bout), h) tongue rolling bout frequency (bout/d), i) tongue rolling total duration (sec/d), j) utilizing brush bout duration (sec/bout), k) utilizing brush bout frequency (bout/d), and l) utilizing brush total duration (sec/d). Different letters indicate statistical differences ($P < 0.05$) among sampling strategies.

## Bar licking

Sampling strategy did not impact the record of average duration of a bar licking bout (Fig 2G), the frequency of these bar licking bouts (Fig 2H), nor the total duration of time spent per day engaged in bar licking behavior (Fig 2I; $P > 0.08$). Bar licking was most accurately captured

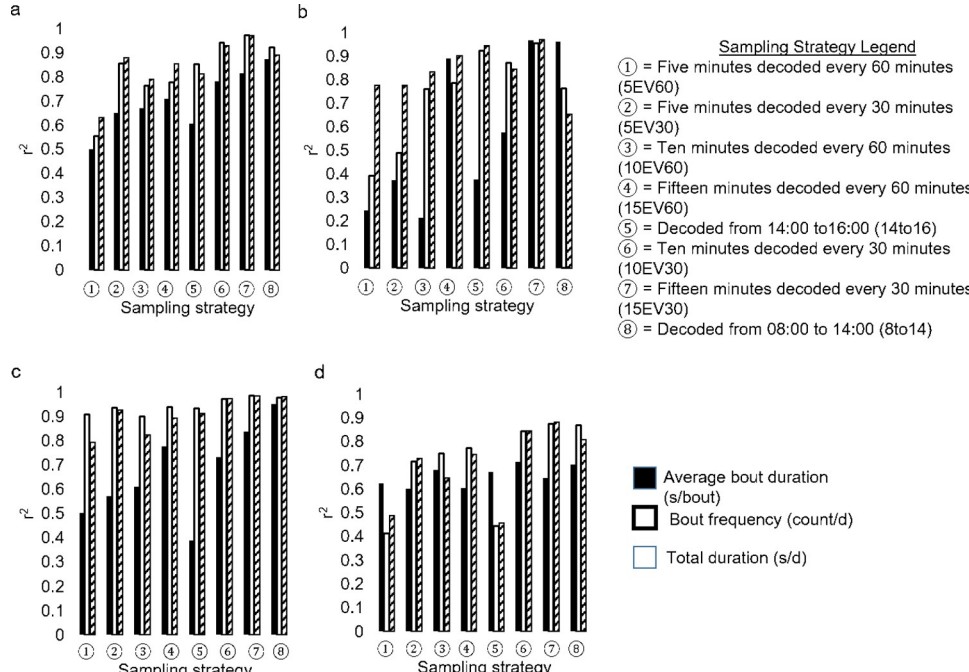

**Fig 3.** Strength of the relationship ($r^2$) between continuous observations (from 8:00 to 20:00) and each sampling strategy for a) allogrooming, b) bar licking, c) tongue rolling, and d) brush utilization for average bout duration (sec), frequency of bout per day, and total duration of all bouts (sec/d). The sampling strategies presented (left to right) are organized from shortest to longest duration of video that would need to be observed to collect the data.

using the 15EV30 sampling strategy ($r^2 > 0.95$; $P < 0.0001$), while the least accurate sampling strategy was 5EV60 ($r^2 > 0.65$; $P < 0.08$) (Fig 3B).

## Tongue rolling

Irrespective of sampling strategy, average bout duration for tongue rolling ($P > 0.99$) did not differ from continuous observations (Fig 2D). The sampling strategies 15EV30 ($P > 0.2$), and 8To14 ($P > 0.5$) did not differ from continuous observations for both tongue rolling frequency (Fig 3E) and the total duration of time per day spent tongue rolling (Fig 2F). The most accurate sampling strategy to capture tongue rolling was 8To14 ($r^2 > 0.95$; $P < 0.0001$), while the least accurate sampling strategy was 14To16 ($r^2 > 0.80$; $P < 0.06$) (Fig 3C).

## Brush utilization

Irrespective of sampling strategy, bout duration for brush utilization ($P > 0.99$) did not differ from continuous observations (Fig 2J). For brush use bout frequency, 15EV30 ($P > 0.2$), and 8To14 ($P > 0.1$) did not differ from continuous observation (Fig 2K). For total duration of brush use per day, 10EV30 ($P > 0.2$), 15EV30 ($P > 0.7$), 14To16 ($P > 0.08$), and 8To14 ($P > 0.1$) did not differ from continuous observations (Fig 2L). The most accurate sampling strategy to capture brush use was 8To14 ($r^2 > 0.70$; $P < 0.0003$), while the least accurate sampling strategy was 14To16 ($r^2 > 0.80$; $P < 0.001$) (Fig 3D).

## Head butt

The sampling strategies 15EV30 ($P > 0.3$) and 8To14 ($P > 0.6$) did not differ from continuous observation for the total number of head butts performed per day within a pen (Fig 4B). The

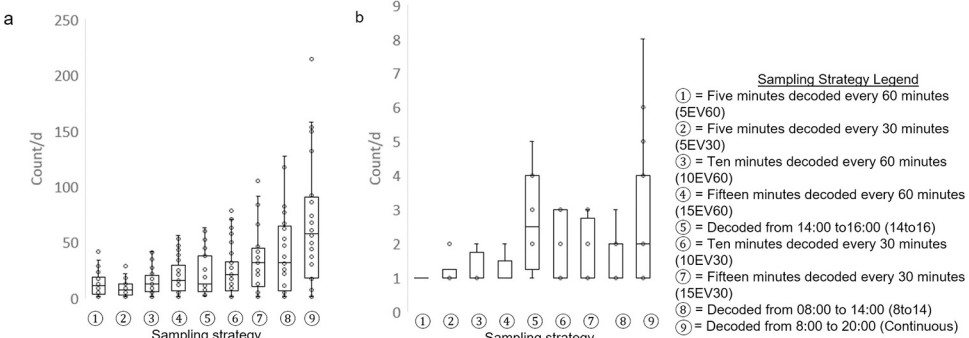

**Fig 4.** The strength of relationship between continuous observations and each sampling strategy when monitoring the daily frequency of drylot-housed steer engaged in a) mounting and b) head butting. Different letters indicate statistical differences ($P < 0.05$) among sampling strategies.

most accurate sampling strategy to capture head butt was 15EV30 ($r^2 > 0.95$; $P < 0.0001$), while the least accurate sampling strategy was 14To16 ($r^2 > 0.80$; $P < 0.001$) (Fig 5B).

## Mounting

Irrespective of sampling strategy, bout frequency for mounting ($P > 0.7$) did not differ from continuous observations (Fig 4A). The most accurate sampling strategy to capture mounting was 15EV30 ($r^2 > 0.75$; $P < 0.05$), while the least accurate sampling strategy was 10EV60 ($r^2 > -0.05$; $P < 0.8$) (Fig 5A).

## Discussion

The objective of this study was to determine which sampling strategies could accurately capture social and stereotypic behavior in cattle housed in feedlots with access to a brush. Observing cattle behavior for 15 minutes every 30 minutes was shown to be the optimal sampling strategy for evaluating daily bout frequency, average bout duration (sec), and total duration of all bouts (sec). This sampling strategy provides insight as to what happens throughout the day and is designed to capture behaviors that happen infrequently or in short durations [15]. On the other hand, observing the animals from 14:00 to 16:00 was shown to be the least accurate

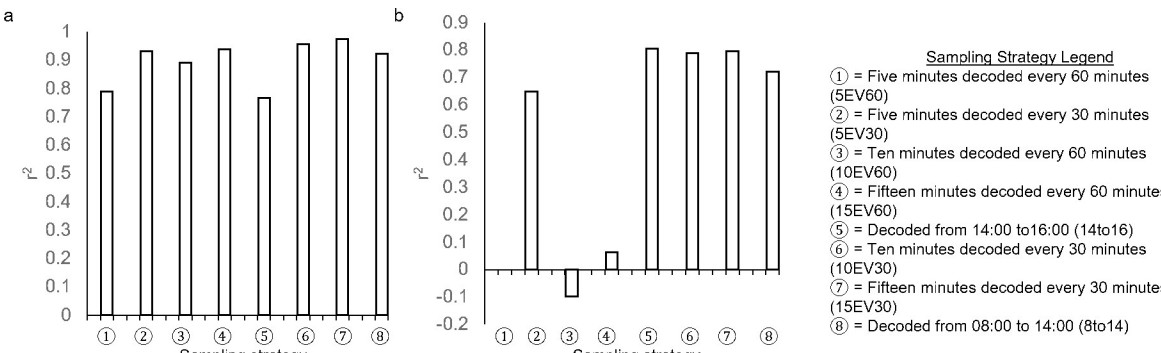

**Fig 5.** The strength of relationship between continuous observations (08:00 to 20:00) and each sampling strategy (indicated in circled numbers) when monitoring the daily frequency of drylot-housed steer a) mounting and b) head butting. The sampling strategies (indicated in circled numbers and described in the legend) presented are organized (left to right) from the shortest to longest duration of time that video recordings would need to be decoded to collect the data.

sampling strategy. That could be due to the circadian pattern of cattle behavior. For example, behaviors such as brush use, mounting, and allogrooming tend to be performed at lower frequencies during the morning than the afternoon [21]. If the time of day can influence when the behaviors are expected to occur, then samplings should be distributed throughout the day; otherwise, the accuracy of the sampling strategy could be negatively impacted.

Contemporary housing systems can result in cattle living in high-density groups where animals may be unable to avoid conflict or may impede upon individual spatial needs. Social animals housed in groups will inevitably develop hierarchies and will engage in social behaviors designed to establish a dominance social structure and dictate resource access [22]. The use of the space varies depending on the animal's dominance status; for example, according to Donaldson et al. [23], low-ranking animals were observed moving constantly to prevent or avoid agonistic behavior. Therefore, being aware of the social structure is crucial to identifying individuals who are experiencing difficulties accessing resources and to identifying any management strategies that can alleviate this social stress and minimize competition for resources.

Mounting was shown to be the most difficult behavior to decode, displaying a negative value in the correlation of 10EV60 with continuous observation, which could be due to the infrequent occurrence of mounting behavior. Feedlot steers housed in pens with a brush perform fewer mounts that feedlot cattle housed without a brush [24]. Mounting is a behavior that has multiple connotations. Mounting is required for copulation, a behavioral signal that can be indicative of reproductive status, and is involved in both affiliative interactions (e.g., play; [24]) and agonistic interactions (e.g., establishing dominance, bulling). Thus, the frequency of this behavior is context specific.

Mounting is the behavior performed during the phenomenon of bulling, a scenario when cattle will perform mounting at an abnormally high rate in which a single individual is the recipient of the mounting attempts, which can result in injury and death. Outbreaks of Buller Steer Syndrome occur in 91% of feedlots with a carrying capacity of over 8,000 and is the third most common health problem (after BRD and digestive problems) for feedlot cattle [25]. Thus, monitoring mounting has welfare and economic implications beyond indicating reproductive status.

Mounting is used to establish dominance, as dominant animals have shown to be responsible for initiating 60% of the mounts in a group. This behavior is also influenced by the novelty of the animals in the group. In pens of newly introduced cattle, animals engage in greater mounting events, suggesting as well that this behavior is involved in social dominance [26, 27], making this behavior complex and variant and therefore hard to measure. Mounting is a behavior that is not only difficult to measure but also difficult to interpret due because of the multidimensional use of this behavioral movement, which can be used for play behavior, social dominance, stereotypic behavior, or reproductive signaling.

Stereotypic behaviors such as bar licking, and tongue rolling can be informative regarding an animal's welfare state. Cattle are orally motivated creatures that evolved to spend most of their days grazing and ruminating [12]. Modern agriculture houses cattle in environments that restrict the performance of or limits the duration of time engaged in these natural behaviors [12]. Environmental enrichment, such as a brush, reduces the performance of tongue rolling and bar licking [14]. Cattle have shown sustained interest in engaging with environmental enrichment like brushes, suggesting that cattle brushes provide mental and physical stimulation. This could be the reason why measuring stereotypic behavior is difficult, as all the pens were equipped with an L-shaped brush, so cattle were not performing stereotypic behaviors frequently.

These findings provide guidance regarding how to expedite large-scale behavior observations that optimize the collection of cattle social behavior data. The present results can be used

to design ways to extend the battery life of sensor technology by generating smaller yet equally informative datasets, thus making data management and output easier to use and resulting in increased adoption and greater economic returns for the producer. This research provides several options, depending on the percentage of accuracy desired by the observer, which could be used in the interest of observing the animal's behavior, optimizing social behavior observation, and offering flexibility to the researcher.

## Supporting information

**S1 Data.**
(XLSX)

## Author Contributions

**Conceptualization:** Claudia Carolina Lozada, Courtney L. Daigle.

**Data curation:** Claudia Carolina Lozada.

**Formal analysis:** Claudia Carolina Lozada, Courtney L. Daigle.

**Funding acquisition:** Courtney L. Daigle.

**Investigation:** Claudia Carolina Lozada.

**Methodology:** Claudia Carolina Lozada, Rachel M. Park, Courtney L. Daigle.

**Project administration:** Courtney L. Daigle.

**Resources:** Rachel M. Park, Courtney L. Daigle.

**Supervision:** Claudia Carolina Lozada, Courtney L. Daigle.

**Validation:** Claudia Carolina Lozada, Courtney L. Daigle.

**Visualization:** Claudia Carolina Lozada.

**Writing – original draft:** Claudia Carolina Lozada.

**Writing – review & editing:** Claudia Carolina Lozada, Rachel M. Park, Courtney L. Daigle.

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
