## [Decision Letter · Decision Letter 0]

2 Aug 2022

PONE-D-22-01864Evaluating accurate and efficient sampling strategies designed to measure social behavior and brush use in drylot housed cattlePLOS ONE

Dear Dr. Daigle,

Thank you for submitting your manuscript to PLOS ONE. After careful consideration, we feel that it has merit but does not fully meet PLOS ONE’s publication criteria as it currently stands. Therefore, we invite you to submit a revised version of the manuscript that addresses the points raised during the review process.

We look forward to receiving your revised manuscript.

Kind regards,

Nei Moreira, Ph.D.

Academic Editor

PLOS ONE

Journal Requirements:

2. Please include a separate caption for each figure in your manuscript.

Reviewers' comments:

Reviewer's Responses to Questions

**Comments to the Author**

1. Is the manuscript technically sound, and do the data support the conclusions?

Reviewer #1: Yes

Reviewer #2: Yes

2. Has the statistical analysis been performed appropriately and rigorously? 

Reviewer #1: Yes

Reviewer #2: Yes

3. Have the authors made all data underlying the findings in their manuscript fully available?

Reviewer #1: No

Reviewer #2: Yes

4. Is the manuscript presented in an intelligible fashion and written in standard English?

Reviewer #1: Yes

Reviewer #2: Yes

5. Review Comments to the Author

Reviewer #1: Many of the title words are found as keywords. Should you choose one or the other?

Evaluating accurate and efficient sampling strategies designed to measure social behavior and brush use in drylot housed cattle

well, which of the two goals is worth?

Thus, the objective of this study was to identify a sampling strategy that could accurately and efficiently record the performance of social, agonistic, and stereotypic behaviors in drylot-housed cattle with access to brushes.

DISCUSSION

The objective of this study was to determine which sampling strategies could accurately capture social and stereotypic behavior in cattle housed in feedlots with access to a brush.

The methodology was previously outlined in Park et al. (2020) ===?? In which of the 2020 works from? There are 3 of them! Didn't show up with a, b, c???

Park, R. M., Cramer, M. C., Wagner, B. K., Turner, P., Moraes, L. E., Viscardi, A. V., ... & Pairis-Garcia, M. D. (2020). A comparison of behavioural methodologies utilised to quantify deviations in piglet behaviour associated with castration. Animal Welfare, 29(3), 285-292.

Park, R. M., Schubach, K. M., Cooke, R. F., Herring, A. D., Jennings, J. S., & Daigle, C. L. (2020). Impact of a cattle brush on feedlot steer behavior, productivity and stress physiology. Applied Animal Behaviour Science, 228, 104995. https://doi.org/10.1016/j.applanim.2020.104995

Park, Rachel M., et al. "Impact of a cattle brush on feedlot steer behavior, productivity and stress physiology." Applied Animal Behaviour Science 228 (2020): 104995.

I think that the work was developed in North America, in the period from December 2017 to June 2018. It needed to have presented the average temperature at which the data set was collected. I needed the rain information. The average gain of the animals and correlated with behavior; temperature; rainfall. The work as it was presented, contributes little to the production system. Even if it's animal welfare.

Reviewer #2: o artigo tem mérito científico, esta bem redigido. Tem dados sólidos e coerentes e pode ser publicado. o material e métodos esta bem descrito. Os resultados obtidos são coerentes, e foram bem discutidos.

6. PLOS authors have the option to publish the peer review history of their article (what does this mean?). If published, this will include your full peer review and any attached files.

Reviewer #1: No

Reviewer #2: No

---

## [Author Response · Author response to Decision Letter 0]

27 Sep 2022

September 9, 2022

Dr. Nei Moreira

Editor

PLOS ONE

Re: Manuscript ID PONE-D-22-01864

Dear Dr. Moreira and anonymous reviewers, 

Thank you for the thorough and thoughtful review of our Manuscript ID PONE-D-22-01864 “Evaluating accurate and efficient sampling strategies designed to measure social behavior and brush use in drylot housed cattle”.

The suggestions made by the reviewers were very helpful to accomplish a more robust and clear paper for the reader. We are grateful for your input and recommendations. Should any part of this manuscript require further alteration, please let us know and we will make the necessary changes.

Below are our responses to the concerns raised by the anonymous reviewers.

Should you have any questions or concerns, please do not hesitate to contact us directly.

Sincerely,

Courtney L. Daigle, Ph.D.

Animal Behavior and Welfare Laboratory

Department of Animal Science

Texas A&M University

College Station, Texas

 

Reviewer #1: Many of the title words are found as keywords. Should you choose one or the other?

Evaluating accurate and efficient sampling strategies designed to measure social behavior and brush use in drylot housed cattle

Response: Thank you for your feedback. New key words were selected, such as environmental enrichment, cattle brush, beef cattle, behavior, and stereotypies.

well, which of the two goals is worth?

Thus, the objective of this study was to identify a sampling strategy that could accurately and efficiently record the performance of social, agonistic, and stereotypic behaviors in drylot-housed cattle with access to brushes.

DISCUSSION

The objective of this study was to determine which sampling strategies could accurately capture social and stereotypic behavior in cattle housed in feedlots with access to a brush.

The methodology was previously outlined in Park et al. (2020) ===?? In which of the 2020 works from? There are 3 of them! Didn't show up with a, b, c???

Park, R. M., Cramer, M. C., Wagner, B. K., Turner, P., Moraes, L. E., Viscardi, A. V., ... & Pairis-Garcia, M. D. (2020). A comparison of behavioural methodologies utilised to quantify deviations in piglet behaviour associated with castration. Animal Welfare, 29(3), 285-292.

Park, R. M., Schubach, K. M., Cooke, R. F., Herring, A. D., Jennings, J. S., & Daigle, C. L. (2020). Impact of a cattle brush on feedlot steer behavior, productivity and stress physiology. Applied Animal Behaviour Science, 228, 104995. https://doi.org/10.1016/j.applanim.2020.104995

Park, Rachel M., et al. "Impact of a cattle brush on feedlot steer behavior, productivity and stress physiology." Applied Animal Behaviour Science 228 (2020): 104995.

Response: I am sorry for the confusion; superscript letters were added to clarify citations. 

I think that the work was developed in North America, in the period from December 2017 to June 2018. It needed to have presented the average temperature at which the data set was collected. I needed the rain information. The average gain of the animals and correlated with behavior; temperature; rainfall. The work as it was presented, contributes little to the production system. Even if it's animal welfare.

Response: A figure with THI, and average precipitations (in) per month in Bushland, TX from November 2017 to July 2018 was added. The goal of this specific paper was not to answer a biological question, per se. The point of this paper was to identify and evaluate different sampling strategies that can be used to collect the social, stereotypic, and environmental enrichment behaviors that would be of interest for a future study.

Reviewer #2: o artigo tem mérito científico, esta bem redigido. Tem dados sólidos e coerentes e pode ser publicado. o material e métodos esta bem descrito. Os resultados obtidos são coerentes, e foram bem discutidos.

Response: Obrigada pela análise e pelos comentários sobre meu artigo.

---

## [Editor Report · Decision Letter 1]

14 Nov 2022

Evaluating Accurate and Efficient Sampling Strategies Designed to Measure Social Behavior and Brush Use in Drylot Housed Cattle

PONE-D-22-01864R1

Dear Dr. Daigle,

We’re pleased to inform you that your manuscript has been judged scientifically suitable for publication and will be formally accepted for publication once it meets all outstanding technical requirements.

Kind regards,

Aziz ur Rahman Muhammad

Academic Editor

PLOS ONE
---

## [Editor Report · Acceptance letter]

23 Dec 2022

PONE-D-22-01864R1 

Evaluating Accurate and Efficient Sampling Strategies Designed to Measure Social Behavior and Brush Use in Drylot Housed Cattle 

Dear Dr. Daigle:

I'm pleased to inform you that your manuscript has been deemed suitable for publication in PLOS ONE. Congratulations! Your manuscript is now with our production department. 

Kind regards, 

on behalf of

Dr. Aziz ur Rahman Muhammad 

Academic Editor

PLOS ONE